# Acceptability, Feasibility, and Appropriateness of Mobile Phone Messaging-Based Message-Framing Intervention for Promoting Maternal and Newborn Care Practices

**DOI:** 10.3390/ijerph22060864

**Published:** 2025-05-31

**Authors:** Hordofa Gutema Abdissa, Gebeyehu Bulcha Duguma, Mulusew Gerbaba, Josef Noll, Demisew Amenu Sori, Zewdie Birhanu Koricha

**Affiliations:** 1Department of Health, Behavior and Society, Faculty of Public Health, Institutes of Health, Jimma University, Jimma P.O. Box 378, Ethiopia; gebeyehubulcha@gmail.com (G.B.D.); zbkoricha@yahoo.com (Z.B.K.); 2Data Science and Evaluation, African Population and Health Research Center, Nairobi 00100, Kenya; mulusew.gerbaba@gmail.com; 3Department of Technology Systems, University of Oslo, 0316 Oslo, Norway; josef.noll@its.uio.no; 4Department of Obstetrics and Gynaecology, Faculty of Medicine, Institutes of Health, Jimma University, Jimma P.O. Box 378, Ethiopia; demisame5@gmail.com

**Keywords:** acceptability, feasibility, appropriateness, mobile phone messaging, SMS, maternal health, newborn health, Ethiopia

## Abstract

There is limited evidence on key implementation outcomes for mHealth interventions that target maternal and newborn health. Hence, this study aimed to evaluate the acceptability, feasibility, and appropriateness of a mobile phone messaging-based message-framing intervention. A cross-sectional study was conducted, involving 397 mothers who participated in the mobile phone messaging-based intervention. Multivariate general linear modeling was carried out to identify factors that were associated with the acceptability, feasibility, and appropriateness of the intervention. The statistical significance level was declared at a 95% confidence interval and *p*-value of <0.05. The mean scores of acceptability, feasibility, and appropriateness were 27.9, 23.8, and 22.5, respectively. Acceptability was significantly affected by living in a rural area, being rich, receiving messages at night, self-efficacy, and engagement. Feasibility was affected by living in rural area, educational status, being a merchant, being rich, receiving messages at night, self-efficacy, engagement, and satisfaction. Meanwhile, appropriateness was influenced by living in a rural area, being a merchant, being a government employee, and satisfaction. The mobile phone messaging-based intervention was highly acceptable, feasible, and appropriate. Focusing on self-efficacy, engagement, satisfaction, the timing for sending messages, and sociodemographic factors would facilitate the implementation and utilization of mobile phone messaging-based interventions.

## 1. Introduction

Globally, an estimated 810 women and over 6700 newborns die every day due to preventable causes related to pregnancy and childbirth. Low- and middle-income countries account for 94% of all maternal and neonatal deaths [1,2,3]. Although there has been substantial improvement in maternal and child survival, mortality among both groups is still unacceptably high [2,4]. In response to this situation, the United Nations has called for action to decrease the maternal mortality ratio (MMR) to below 70/100,000 live births and neonatal mortality to below 12 deaths per 1000 live births [5]. The risk of death due to complications during pregnancy, labor, delivery, and the postnatal period is assumed to be reduced through the provision of antenatal care (ANC), institutional delivery, and postnatal care services. However, low utilization of these services remains a challenge globally, particularly in developing countries [6].

Ethiopia has been implementing programs and strategies such as emergency obstetric care, emergency newborn care, the obstetric and gynecologic problem referral and network system, maternity waiting homes, community-based newborn care, integrated management of newborn and childhood illness, the integrated community case management, the newborn care corner initiative, neonatal intensive care units (NICUs), and pediatric referral care to improve maternal and neonatal morality [7,8]. However, the country is still among the top four countries with high neonatal mortality, alongside India, Nigeria, and Pakistan. It is also among 15 countries with a high MMR [1,3]. Hence, new interventions are needed to improve maternal and newborn health and to support further government efforts to reduce the MMR and neonatal mortality rates. A recent systematic review suggests that providing community-based educational interventions during antenatal and postnatal periods can reduce overall neonatal and perinatal mortality and late neonatal mortality [9]. 

Digital health is a prominent practice involving the use of information and communications technology to overcome health issues [10]. The growth of mobile connectivity created an opportunity for the birth of a digital health component called mHealth [11]. More specifically, mHealth is a health practice that uses the short message service (SMS), voice messages, text, audio, video, images, and applications [12]. mHealth has the ability to overcome barriers to healthcare access, such as geographic distance from services, social marginalization, a lack of skilled medical personnel, or a lack of financial resources, which are responsible for high maternal and neonatal mortality in developing countries [13,14]. Mobile phone messaging or SMS is a component of mHealth that has shown promising outcomes in regard to promoting maternal and child healthcare services [15,16,17,18] and was found to be an acceptable means of educating mothers about maternal and newborn care [19,20]. 

Message framing (MF) is a communication strategy that aims to influence behavior by presenting and structuring messages in the form of the benefits of practicing a particular behavior (i.e., gain-framed) or the consequences of failing to practice that behavior (i.e., loss-framed) [21]. MF is inspired by prospect theory, which assumes that gain-framed messages (GFMs) are more effective if people perceive that the behavior stated in the message indicates a potential benefit, and that loss-framed messages (LFMs) are more effective if people perceive that the behavior stated in the message has a potential risk [22]. It has been suggested that GFMs are more persuasive than LFMs for promoting disease prevention behaviors [23]. However, reports on the effectiveness of GFMs vs. LFMs in promoting health behavior have shown inconsistent findings, according to which neither GFMs nor LFMs promoted the desired behavior [24], no significant difference was observed between the effect of GFMs and LFMs [25,26], and LFMs were found to be more effective than gain-framed ones [27]. Moreover, the evidence of MF effects on maternal and newborn care practices is still unclear in LMICs [28]. Therefore, understanding which type of MF has a more persuasive effect in regard to maternal and newborn care practices along with the implementation outcomes in the Ethiopian setting needs further confirmation. 

The end-line assessment that we conducted in January 2024 evaluated the effect of a gain- vs. loss-framed intervention delivered through mobile phone messaging for promoting maternal and newborn care practices among mothers [29]. Following this end-line evaluation, we assessed the implementation outcomes, considering that they would help us interpret the effect of the intervention on promoting maternal and newborn care practices and identify the main facilitators and barriers to the intervention [30]. Assessing implementation outcomes along with effectiveness can speed up dissemination and translation of the research into usual practice by measuring how well the intervention can be implemented in real-world settings [31]. Acceptability, feasibility, and appropriateness are the most prominent measures [30], and perceptual implementation outcomes that are proximal can be strongly associated with the effectiveness of the intervention [32]. Scholars have assessed the acceptability, feasibility, and appropriateness of several health interventions targeting different implementation stakeholders [33,34,35,36]. However, studies that assessed these implementation outcomes for mobile phone messaging-based intervention reported varied measurements [37], while others used only a few or one item to report them [35,38,39]. Moreover, these studies fail to assess whether the end-users received the intervention according to their preference, participant level of exposure, engagement, and satisfaction with the intervention.

This study desired to answer the following questions: (1) What is the level of acceptability, feasibility, and appropriateness of mobile phone messaging-based interventions? (2) What is the level of end-user exposure, engagement, and satisfaction with mobile phone messaging-based interventions? (3) What factors affect mobile phone messaging-based intervention acceptability, feasibility, and appropriateness? Thus, focusing on the end-user perspective, the principal aim of this study is to assess acceptability, feasibility, and appropriateness of a mobile phone messaging-based message-framing intervention for promoting maternal and newborn care practice. 

## 2. Materials and Methods

### 2.1. Study Design 

A cross-sectional study was conducted in February 2024 a month after end-line assessment of an intervention that used a three-armed cluster randomized control trial design. This intervention was implemented from May 2023 to December 2023 and the end-line assessment was conducted in January 2024 [29].

### 2.2. Population and Sample

This study was conducted among mothers who were recruited and participated in the intervention arm of the trial. The trial randomly assigned 21 clusters into 3 arms: (1) gain-framed arm, (2) loss-framed arm, and (3) control arm. The total sample size of the trial was 588, and 196 pregnant mothers were recruited for each arm. The detailed description of the sample size of the trial is presented elsewhere [29]. The current study considered mothers who took the intervention package, namely those recruited to the gain- and loss-framed arms. However, since the population of these two groups is small, we included all mothers in these groups without sample size calculation. Accordingly, the sample size for the current study was 392 mothers. This study was conducted in the Manna, Shebe-Sombo, and Dedo districts of Jimma Zone, Oromia, Ethiopia. Jimma Zone has poor health infrastructure, low health worker density (3.04 health workers per 1000 population), few health facilities that are offering Comprehensive Emergency Obstetric and Newborn Care.

### 2.3. Inclusion and Exclusion Criteria

All mothers who took part in the intervention and received at least one intervention text message through their or a family member’s phone were included in this study. Mothers who were identified as lost to follow-up due to death or relocated from the study area during the study period, or who were unable to communicate due to severe illness, were excluded from this study.

### 2.4. Description and Approach of the Intervention

The intervention was designed based on findings from its formative and development phase. The formative assessment explored facilitators and barriers to using mobile phone text messages to deliver education on maternal and newborn care. This formative assessment also identified pregnant women’s experience, intention to use, and perceived acceptability of mobile phone messages as a means to receive information on maternal and newborn care. Following the formative assessment, intervention messages were developed in the form of gain- and loss-framed formats based on the concept of message framing. The gain-framed messages emphasized the benefit of maternal and newborn healthcare practices, while the loss-frame messages emphasized the risk or loss as a result of not utilizing maternal health services and practicing newborn care. 

We developed and refined the messages in several phases, engaging various stakeholders. The initial 120 messages were drafted based on information provided to pregnant women during counselling in pregnancy, childbirth, and after childbirth as presented in Ethiopian Ministry of Health guidelines [40] and World Health Organization recommendations [41]. After discussing with the research team, 32 messages were deleted. Then, the remaining 88 messages were translated into Afan Oromo language (the local language) and were evaluated by mothers and experts before being used in the intervention. A closed card-sorting activity was conducted with 12 pregnant mothers to determine how they understood each message. The card containing the Afan Oromoo version of the selected message was given to the mothers. The mothers sorted the messages into predefined piles that were labeled with different thematic categories. 

Data from this card-sorting activity were entered into an adapted Excel 2019 spreadsheet template developed to analyze card-level agreement with the original thematic area [42]. Messages that > 50% of participants sorted into the intended thematic area were considered for the next phase. During this card-sorting activity, ten messages were excluded as they did not meet the above criteria. Subsequently, eight experts assessed the reserved 78 messages for clarity, ease, tone, believability, benefit, receiving attention, appropriate of language, and potential offensiveness. This assessment was conducted using ten five-point Likert-scale items that were developed based on the health communication messages review criteria [43]. Fourteen messages (thirteen loss-framed and one gain-framed) were modified based on the experts’ suggestions. Finally, 78 messages were retained—39 gain-framed and 39 loss-framed, organized into ten thematic areas. 

A system that manages the schedule and sends the messages automatically was developed locally. A domain, layer 3 virtual private network, and short code were purchased from Ethio-telecom company. The shortcode was named “Ulfa Mijuu”, which means “Safe Pregnancy” in Afan Oromoo. The software(Ulfa mijuu version 1.02) interface allowed us to enter the participant’s name, gestational age, mobile number, message type, category, language option, and report generation and feedback option.

Participants in the gain-framed arm received weekly automated gain-framed messages from enrolment to the postnatal period. Likewise, participants in the loss-framed arm received weekly loss-framed messages over the same period. All the messages the two groups received were tailored and scheduled based on each woman’s gestational age, expected delivery date, and postnatal period. Personalized maternal and newborn health-related text messages were dispatched to the participant’s registered phone number. Participants in the control arm received existing antenatal care (ANC) and postnatal care (PNC) services provided at health facilities. Details about the intervention phases and the implementation process were presented in the previously published study [29].

### 2.5. Data Collection Tool and Procedure

Data were collected using a structured, interviewer-administered questionnaire adapted from related studies [30,34,35,36,44]. The questionnaire was developed in English and translated into Afan Oromo and back-translated into English to ensure consistency. Six data collectors and two supervisors were recruited and received two-day training on the study’s objective, questionnaire content, measurements, ethical considerations, and use of the Open Data Kit (ODK-2.0) application. Before the main data collection, a pretest was conducted on 5% of the sample size in a kebele which is outside of the study area, and modifications were made to the questionnaire based on the results. Data completeness was checked during data collection by supervisors, and any questionnaires with missing items were submitted to data collectors for correction.

### 2.6. Study Variables and Measurements

Acceptability, feasibility, and appropriateness were the dependent variables in this study. The independent variables were sociodemographic characteristics (age, educational status, ethnicity, religion, marital status, occupational status, residence, education of husband, occupation of husband, monthly income, family member able to read text messages), exposure to intervention (the number of text message received, who read the message, message category, frequency, and timing of the received message), self-efficacy, satisfaction with the intervention, and engagement with the intervention. The constructs were operationalized and measured as follows.

Acceptability: Defined as the perception of mothers on receiving maternal and newborn health information via mobile phone messages as beneficial or satisfactory to them in fulfilling their needs [30,35]. Eight items were used on a five-point Likert scale ranging from 1 = strongly disagree to 5 = strongly agree [34]. Item scores were added after reverse coding negatively worded items to create a composite variable scale. The internal consistency of acceptability items was scored with a Cronbach’s alpha coefficient of 0.81. A high score indicated greater acceptability of the mobile phone messaging-based message-framing intervention.

Feasibility: Defined as the extent to which the mobile phone messaging-based message-framing intervention was perceived as successfully promoting maternal and newborn care practice within their setting [30]. It was measured using six 5-point Likert scale items. Item scores were added after reverse coding negatively worded items to create a composite variable scale. The internal consistency of the items was scored with a Cronbach’s alpha coefficient of 0.82. A higher score implied that mobile phone messaging-based message framing is a feasible strategy for promoting maternal and newborn practice. 

Appropriateness: Defined as the mother’s perception of the fit, relevance, and compatibility of the mobile phone messaging-based message-framing intervention as a means of communicating about maternal and newborn care in their setting [30,35]. Five 5-point Likert scale items were used and item scores were added after reverse coding negatively worded items to create a composite variable scale. The internal consistency of the items was Cronbach’s alpha coefficient of 0.77. A higher score implied that mobile phone messaging-based message framing is an appropriate strategy in promoting maternal and newborn practice.

Self-efficacy of utilizing mobile phone messaging: Defined as a mother’s belief in her ability to use a mobile phone for text messages. It was measured using five 5-point Likert scales to measure this component, in which sum scores were used in the analysis after reverse coding negatively worded items. The internal consistency of self-efficacy items was scored with a Cronbach’s alpha coefficient of 0.89. A score with a high value indicated higher self-efficacy toward SMS utilization.

Satisfaction: Defined as the extent to which mothers judge and perceive the outcomes or experience of using mobile phone messaging-based message-framing intervention. It was measured using seven 5-point Likert scale items and the respondents were asked to rate to what extent they agreed on each item. Item scores were added after reversely coding negatively worded items to create a composite variable scale. The internal consistency of satisfaction items was scored with a Cronbach’s alpha coefficient of 0.91. A higher rating corresponds to higher satisfaction with the mobile phone messaging-based message-framing intervention.

Engagement: Defined as a mother’s experience, focusing on their involvement and interaction with the mobile phone messaging-based message-framing intervention. It was measured using seven five-point Likert scale items, and item scores were added after reversely coding negatively worded items to create a composite variable scale. The internal consistency of the items was scored with a Cronbach’s alpha coefficient of 0.9. A higher rating corresponds to a higher engagement with the mobile phone messaging-based message-framing intervention [45].

### 2.7. Data Quality and Management

The face and content validity of the tool were ensured by adapting the tool from previous related studies. Prior to the main data collection, the questionnaire was pretested on 5% of the tool sample size in a cluster that was not included in the intervention, and modifications were made to the questionnaire based on the findings. The internal consistency of the key constructs was checked using Cronbach’s alpha, and all the constructs showed acceptable values (≥0.7) of Cronbach’s alpha coefficient.

### 2.8. Ethics

Ethical approval was obtained from the Jimma University Institutional Review Board (ref no. JUIH/IRB/358/23). This study was conducted in accordance with the guideline of the Declaration of Helsinki. Informed consent was obtained from all participants. Participants were informed the purpose of this study, the researchers and institutions involved, the expectations of women, and potential risks and benefits associated with this study. They were also informed of their rights as participants and had their questions answered before enrollment. Confidentiality and anonymity were assured throughout data collection and analysis.

### 2.9. Data Analysis

The data were analyzed using the Statistical Package for Social Sciences (SPSS) version 27. Descriptive statistical measures such as frequency, mean, proportion, and standard deviation were computed and presented using tables to summarize the data. The normality of implementation outcomes and other continuous variables was checked using the Shapiro–Wilk statistical test, appropriate for our small sample size and graphical methods, namely Normal Q-Q plots and histograms prior to conducting any inferential analysis. There were minor violations of the normality assumption. Considering this minor deviation from normality and the robustness of this test, particularly for the smaller sample size of 379 in this study, Pearson’s correlation analysis was carried out to examine the linear relationship between the implementation outcomes and other psychometric constructs as a bivariate analysis. Moreover, an independent *t*-test and one-way ANOVA were conducted to compare the means of continuous variables between two independent groups and among three or more independent groups, respectively. 

Multivariate general linear modeling (GLM) was used to evaluate the relationship between acceptability, feasibility, and appropriateness (dependent variables) and the predictors. The normality of the residuals for each dependent variable in the GLM was assessed using the Shapiro–Wilk test and graphical methods such as Normal Q-Q plots and histograms. Acknowledging a minor violation of the normality of residuals for the dependent variables, we proceeded with GLM given the robustness of the test and the sample size. The model included all relevant predictors, and regression coefficients (beta, β) from the multivariate linear model were used to interpret the association between dependent and independent variables. Statistical significance of all analyses was determined at the 95% confidence interval, with a *p*-value < 0.05 considered statistically significant.

## 3. Results

### 3.1. Participant Background Characteristics

A total of 379 mothers participated in this study, forming a response rate of 96.7%. Of the participants considered to take part in this study, 13 of them did not participate due to maternal death (*N* = 3) or a change in residential area (*N* = 10). The mean age of participants was 27.24 ± SD 3.98 years. Over two-thirds (67.8%) of the participants resided in rural areas. The vast majority of them were Oromo ethnicity (93.4%) and Muslim by religion (89.7%). Three hundred and seventy-two (98.2%) of the participants were married. In terms of educational status, 45.6% of participants attended primary school, while more than half (56.7%) identified as housewives. Regarding wealth status, 30.4% of the participants were in the richest wealth category, followed by medium (27.7%) (See Table 1).

### 3.2. Reach, Exposure, and Engagement with Ulfa Mijuu Text Messages

The number of Ulfa Mijuu (UM) text messages sent to participants was obtained from the automated software from which they were dispatched. Overall, 15,288 messages were planned to be sent to participants, and 15,132 (98.9%) messages were successfully dispatched. The reach of UM messages was measured based on the number of read messages. Accordingly, more than half of the participants (54.1%) read all of the messages they received, while 120 (31.7%) of them read most of the messages. Regarding recall of UM messages’ exposure, 58.3% of the participants received messages weekly. The higher exposure recall was for ANC message content (87.1%), and the lowest was for lifestyle modification message (64.6%) content. Two-thirds (63.6%) of the participants had a high level of engagement with the UM messages. Most of the participants received the messages on their phones (73.1%) and read them themselves (68.6%). One hundred and thirty-three (35.1%) participants received UM messages in the morning, and the majority of the participants (82.3%) stated they received them at their preferred time (see Table 2).

### 3.3. Descriptive Statistics and Internal Consistency of the Constructs

The mean score for acceptability was 27.9 (SD = 4.7, range = 7–35). The mean scores of feasibility and appropriateness were 23.8 (SD = 4.3, range = 6–30) and 22.5 (SD = 4, range = 5–25), respectively. Self-efficacy was found to have a mean score of 18.7 (SD = 5, range = 5–25). Satisfaction and engagement had mean scores of 27.01 (SD = 4.9, range = 7–35) and 26.5 (SD = 5.4, range = 7–35), respectively.

The internal consistency of the constructs was checked using Cronbach’s alpha coefficient. All the key constructs showed an acceptable level of reliability, with all items showing high item–total correlation. Among the constructs, satisfaction had a higher alpha coefficient score (α = 0.91) when compared to the other constructs. The appropriateness construct had a lower alpha coefficient (α = 0.77), but all the items had better item–total correlations. The alpha coefficient score of acceptability was 0.81, after deleting one item due to a lower item–total correlation. The alpha coefficient score of feasibility was 0.82, following removal of one item with a lower item–total correlation. The alpha coefficient score of self-efficacy was 0.89, after deleting two items from this dimension for having lower item–total correlations. Engagement was found to have the second highest alpha coefficient score of α = 0.9 after three items with lower item–total correlations were removed (see Table 3).

### 3.4. Descriptive Statistics and Pearson Correlation 

A correlation analysis was conducted to examine the relationships between implementation outcomes and other psychometric measurements. Pearson’s correlation coefficients (r) revealed that all measurement scales were positively and significantly correlated with different levels of strength. The highest strong positive significant correlation was observed between satisfaction and engagement (r = 0.792, *p* < 0.01), while a weak positive correlation was observed between acceptability and appropriateness (r = 0.334, *p* < 0.01). Acceptability showed strong positive correlations with feasibility (r = 0.649, *p* < 0.01), self-efficacy (r = 0.624, *p* < 0.01), and engagement (r = 0.608, *p* < 0.01) and a moderate positive correlation with satisfaction (r = 0.577, *p* < 0.01). 

Feasibility also had a strong positive correlation with self-efficacy (r = 0.605, *p* < 0.01), satisfaction (r = 0.678, *p* < 0.01), and engagement (r = 0.688, *p* < 0.01). It had a moderate positive correlation with appropriateness (r = 0.513, *p* < 0.01). Appropriateness had moderate positive correlations with self-efficacy (r = 0.437, *p* < 0.01), satisfaction (r = 0.577, *p* < 0.01), and engagement (r = 0.475, *p* < 0.01). Self-efficacy had a strong positive correlation with both satisfaction (r = 0.647, *p* < 0.01) and engagement (r = 0.684, *p* < 0.01) (see Table 4).

### 3.5. Predictors of Acceptability, Feasibility, and Appropriateness 

A multivariate general linear model was used to identify the association between the three dependent variables (acceptability, feasibility, and appropriateness) and independent variables significantly associated with all or at least one of the dependent variables in bivariate analysis. The bivariate analysis showed that residence, occupation, wealth index, exposure to the messages, satisfaction, self-efficacy, and engagement were significantly associated with all three dependent variables. Additionally, educational status, owner of the phone used to receive the messages, the person reading the message, timing of message receipt, and receiving the message at the preferred time were significantly associated with at least two of the dependent variables. However, in the multivariate general linear model, only residence, educational status, occupation, satisfaction, self-efficacy, engagement, and message receipt time had a significant association with at least one of the dependent variables. 

Residence, wealth index, UM messages’ receipt time, self-efficacy, and engagement with the intervention were predictors of the acceptability of the intervention. Accordingly, living in rural areas (β = −1.827, *p*-value < 0.002) was negatively associated with acceptability. This means that the acceptability of the intervention was, on average, 1.827 units lower among those living in rural areas than urban residents. Participants categorized as rich had a 2.2 (β = 2.193, *p*-value < 0.01) times higher perception of the acceptability compared to their counterparts. Receiving UM messages in the evening was negatively associated with the acceptability of the intervention (β = −1.218, *p*-value < 0.007), indicating lower acceptability of the intervention among those who received UM messages in the evening compared to those who received them in the morning. Additionally, self-efficacy (β = 0.354, *p*-value < 0.000) and engagement (β = 0.202, *p*-value < 0.001) were positively associated with the acceptability of the intervention. This implies that more belief in participants’ confidence to use UM messages and a higher engagement with the intervention are linked to an increased acceptability of the intervention. 

The feasibility of the intervention was predicted by residence, educational status, occupation, wealth index, UM messages’ receipt time, self-efficacy, satisfaction, and engagement. Feasibility of the intervention was decreased by 1.885 units (β = −1.885, *p*-value < 0.001) among participants who were living in rural areas compared with those who were living in urban areas. Feasibility of the intervention was positively associated with educational status, where it was 1.47 units (β = 1.47, *p*-value < 0.018) higher among participants who could read and write than those who were unable to read and write. Moreover, the feasibility of the intervention was increased by 0.975 units (β = 0.975, *p*-value < 0.02) among merchants compared to housewives. Those who were in the rich category of the wealth index had a 1.44 (β = 1.439, *p*-value < 0.042) times higher perception of the feasibility of the intervention compared to their counterparts. Feasibility of the intervention was decreased by 1.097 units (β = −1.097, *p*-value < 0.003) among those who received UM messages in the evening compared to those who received them in the morning. Feasibility of the intervention also had a positive significant association with self-efficacy (β = 0.132, *p*-value < 0.003), engagement (β = 0.251, *p*-value < 0.000), and satisfaction (β = 0.183, *p*-value < 0.001).

Appropriateness of the intervention was significantly associated with residence, occupation, and satisfaction with the intervention. Appropriateness of the intervention decreased by 1.355 units (β = −1.355, *p*-value < 0.005) among participants who were living in rural areas compared with urban residents. Appropriateness of the intervention was 3.727, 4.086, and 2.477 units higher among merchants (β = 3.727, *p*-value < 0.000), government employees (β = 4.086, *p*-value < 0.000) and individuals in other occupation categories (students and daily laborers) (β = 2.477, *p*-value < 0.002), respectively, compared to those who were housewives. Satisfaction with the intervention was also positively associated with the appropriateness of the intervention (β = 0.256, *p*-value < 0.000) (see Table 5).

## 4. Discussion

This study sought to address the gaps in the literature regarding implementation outcome assessment. It examined participants’ levels of exposure and whether their preferences in regard to receiving the intervention were fulfilled in the current study in addition to assessing the acceptability, feasibility, and appropriateness of the message-framing intervention delivered via mobile phone messaging. This study has also contributed to the existing body of knowledge, especially in developing countries, where there is a scarcity of evidence on the subject matter. Exposure to UM messages refers to how frequently participants received them and the proportion of exposure to the message type. Although UM messages were sent to participants every week, only three in five of them reported they had received the messages weekly. Some participants may have missed weekly messages due to poor network coverage or electric power outage. This challenge might have been resolved if the system sending the messages provided a delivery report and automatically resent undelivered messages. 

Participants who received UM messages via family members’ phones might mainly have missed the weekly message since 43.5% did not regularly have access to the phone the phone and, in 22.4% of cases, family members forgot to inform them about the messages. More than half (54.1%) of the participants read all of the messages they received, while 31.7% of them read most of the messages, and this implies that UM messages achieved high reach. ANC-related messages had a higher exposure recall (87.1%), while the lower (64.6%) exposure recall was related to lifestyle message content. Regarding engagement with the intervention, 63.6% of the participants had high levels of engagement with the UM messages. High engagement with an intervention has been reported by other studies that used SMS to increase adherence to triage among HPV-positive women [35] and to educate mothers caring for Aboriginal and Torres Strait Islander children aged <5 years [46]. 

Our study showed high mean scores for acceptability, feasibility, and appropriateness. The literature also presents interventions that had high scores of acceptability, feasibility, and appropriateness [34,47]. These high scores imply that the interventions delivered through mobile phone texting were well-received, practical, and a good fit for educating mothers on maternal and newborn health issues in the study area. The high acceptability in the current study is consistent with other studies that showed high acceptability of SMS-based interventions [33,35]. This high figure might be related to engaging experts and target groups in the message development process. The timing of the messages sent based on the participants’ preferred times could be another reason for the high acceptability of the intervention. The high mean scores of feasibility and appropriateness reported in our study were also consistent with previous studies [35,36]. The observed higher feasibility and appropriateness in our study might be due to the personalization of the messages, the frequency, and the number of messages used in the intervention. Assistance provided from community volunteers to participants who experienced difficulty in using the messages may have also increased the feasibility and appropriateness in our study. 

In this study, internal consistency was assessed by computing Cronbach’s alpha as a psychometric measurement of the implementation outcomes. Accordingly, acceptability, feasibility, and appropriateness were found to have strong internal consistency. This finding was comparable with other studies, where acceptability [33,34], feasibility, and appropriateness [33,34,36] had high internal consistency. Moreover, the current study revealed that acceptability, feasibility, and appropriateness were distinct but correlated implementation outcomes. Acceptability was found to have a strong correlation with feasibility. This is also supported by previous studies that reported high [30,34] and moderate correlations [33,47] between acceptability and feasibility, while acceptability is weakly correlated with appropriateness. Studies showed a high [34,36] to moderate correlation [33] of acceptability and appropriateness. These correlations imply mothers are more likely to use mHealth interventions they perceive as a good fit to their situation (appropriate) and practical to use in their setting (feasible), which are regarded as more acceptable. Feasibility and appropriateness were moderately correlated in our study. However, previous studies have shown a high correlation between feasibility and appropriateness [33,34]. 

This study also identified factors affecting the acceptability, feasibility, and appropriateness of SMS-based interventions. Self-efficacy had a positive association with both the acceptability and feasibility of the intervention. This aligns with previous study findings that showed confidence in one’s ability to engage in an intervention positively affected its acceptability and feasibility [47]. Participant engagement also positively influenced the intervention’s acceptability and feasibility. Similar results have been reported in the literature where user engagement has been shown to increase the acceptability of digital health interventions [46]. Satisfaction with SMS-based maternal education was also positively associated with the feasibility and appropriateness of the intervention. Tailored messages delivered at specific stages of pregnancy to provide timely information on pregnancy, childbirth, and newborn care may have contributed to increased mother satisfaction and increased perceptions of the intervention’s practicality and appropriateness. Receiving UM messages in the evening negatively affected both the acceptability and feasibility of the intervention in our study. In contrast, another study that assessed the influence of content relevance and delivery time on SMS showed a higher acceptance of SMS in the afternoon and evening than in the morning [48]. Differences in context, study objectives, and participant characteristics may be reasons for the discrepancy between the findings. 

Moreover, this study identified several sociodemographic factors that influenced implementation outcomes. Living in a rural area was found to negatively affect the acceptability, feasibility, and appropriateness of the intervention. Participants from rural areas may have encountered limited mobile network coverage, which potentially reduced their satisfaction and led to negative perceptions of the intervention’s practicality and appropriateness. This finding implies the need to consider participants’ places of residence when designing and implementing an mHealth intervention. This study also revealed the effect of wealth on implementation outcomes. Participants with higher income levels were more likely to find the intervention acceptable and feasible. Although not directly linked to implementation outcomes, previous studies have shown a high uptake of digital health interventions among individuals with high incomes [49]. 

Being a merchant was positively associated with the feasibility of the intervention. Merchants, government employees, daily laborers, and students perceived the appropriateness of the intervention as greater than housewives did. Merchants, government employees, and students may perceive the intervention as practical and appropriate as they often have better access to and familiarity with technology and are more frequent users of SMS. Participants who could read and write considered the intervention more feasible than those who could not read or write. Previous studies have also shown a positive association between educational status and implementation outcomes [50,51]. This implies that a higher level of education may improve the understanding of digital health programs, their use, and how to address challenges associated to digital health interventions. Therefore, taking the educational status into account when designing and implementing SMS-based maternal education is crucial to ensure its practicability. 

The high levels of acceptability, feasibility, appropriateness, and engagement observed in this study indicate that such an intervention can effectively deliver important maternal and child health information, particularly to mothers with limited access to health facilities. Future programs should prioritize the observed disparities in access to this type of intervention. The study also identified that sharing mobile phones with family members affected access to the messages. This implies the need to find options for delivering messages to multiple phone numbers to ensure they reach the intended audience. Engaging community volunteers or other supportive staff with mHealth interventions could help overcome the challenges end-users face in accessing and using the service. 

Our study has the following limitations. First, it was at risk of social desirability bias. As we reported the findings based on self-reported data, they may have been subject to social desirability bias. Recall bias might have affected the study measurements, particularly in measuring the exposure to UM message-related variables. Additionally, psychometric properties such as test–retest reliability and discriminant and convergent validity were not assessed in this study. Finally, this study assessed the implementation outcomes only from the end-users’ perspective, as other stakeholders, such as service providers or policymakers, were not involved in the intervention. Therefore, it might not reflect other stakeholders’ views.

## 5. Conclusions

The current study identified high acceptability, feasibility, and appropriateness, indicating that a mobile phone messaging-based intervention was appealing, practical, and a good fit for educating mothers on maternal and newborn healthcare. A message-framing intervention delivered via mobile phone text messages, focused on self-efficacy, user engagement, satisfaction, timing for sending messages, and key sociodemographic factors such as residential area, wealth, and educational status, may facilitate the implementation and utilization of such program. Future studies should explore other implementation outcomes and engage different stakeholders.

## Figures and Tables

**Table 1 ijerph-22-00864-t001:** Participant sociodemographic characteristics from mobile phone messaging-based message-framing intervention, Jimma, Ethiopia (*N* = 379).

Variables		Frequency	Percent
Age	15–19	5	1.3
	20–24	68	17.9
	25–29	198	52.2
	30–34	79	20.8
	35–39	28	7.4
	40–44	1	0.3
Residence	Rural	257	67.8
	Urban	122	32.2
Ethnicity	Oromo	354	93.4
	Others ^a^	25	6.6
Religion	Muslim	340	89.7
	Catholic	23	6.1
	Others ^b^	16	4.2
Marital status	Married	372	98.2
	Others ^c^	7	1.9
Educational status	Can not read and write	77	20.3
	Can read and write	36	9.5
	Primary school	173	45.6
	Secondary school	53	14
	Collage and above	40	10.6
Occupation	Housewife	215	56.7
	Merchant	80	21.1
	Government employee	62	16.4
	Others ^d^	22	5.8
Wealth index	Poorest	75	19.8
	Poor	57	15
	Medium	105	27.7
	Rich	27	7.1
	Richest	115	30.4

^a^ (Amhara, Yem, and Kefa), ^b^ (Orthodox and Protestant), ^c^ (Single, divorce, and widowed), ^d^ (Student and daily laborer).

**Table 2 ijerph-22-00864-t002:** Exposure, reach, and engagement with mobile phone messaging-based message-framing intervention, Jimma, Ethiopia (*N* = 379).

Variables		Frequency	Percent
Frequency of UM message received	Monthly	40	10.6
	Twice a month	118	31.1
	Weakly	221	58.3
Exposure to UM message contents	Antenatal care	330	87.1
	Pregnancy/delivery danger sign	297	78.4
	Necessity of birth planning	294	77.6
	Child immunization	282	74.4
	Postnatal care	267	70.4
	Newborn care	279	73.6
	Benefit of breast feeding	282	74.4
	Maternal and newborn nutrition	280	73.9
	Newborn danger sign	277	73.1
	Personal Hygiene	260	68.6
	Improving lifestyle	245	64.6
Read UM messages	Few of the messages	54	14.2
	Most of the messages	120	31.7
	All of the messages	205	54.1
Level of engagement	Low engagement	138	36.4
	High engagement	241	63.6
Owner of phone used to receive UM messages	Self	277	73.1
	Family member	102	26.9
Who read the UM messages	Self	260	68.6
	Family member	119	31.4
Problem faced to receive UM message on others’ phone (*N* = 102)	No problem faced	37	37.7
	No access to the phone all the time	43	43.9
	Family member forgets informing	22	22.4
UM messages receipt time	Morning	133	35.1
	Afternoon	117	30.9
	Evening	129	34
UM messages received at preferred time	Yes	312	82.3
	No	67	17.7

**Table 3 ijerph-22-00864-t003:** Summary and internal consistency measures for acceptability, feasibility, and appropriateness of mobile phone messaging-based message-framing intervention, Jimma, Ethiopia (*N* = 379).

Variable	Mean	Std. Dev	Alpha
Acceptability	27.9	4.7	0.81
Feasibility	23.8	4.3	0.82
Appropriateness	22.5	4	0.77
Self-efficacy	18.7	5	0.89
Satisfaction	27.01	4.9	0.91
Engagement	26.5	5.4	0.9

**Table 4 ijerph-22-00864-t004:** Descriptive statistics and Pearson’s correlation coefficients measure of acceptability, feasibility, and appropriateness of mobile phone messaging-based message-framing intervention, Jimma, Ethiopia (*N* = 379).

Variables		1	2	3	4	5	6
1	Acceptability	-					
2	Feasibility	0.649 **	-				
3	Appropriateness	0.334 **	0.513 **	-			
4	Self-efficacy	0.624 **	0.605 **	0.437 **	-		
5	Satisfaction	0.577 **	0.678 **	0.577 **	0.647 **	-	
6	Engagement	0.608 **	0.688 **	0.475 **	0.684 **	0.792 **	-
Number of items		7	6	5	5	7	7
Range		7–35	6–30	5–25	5–20	7–35	7–35

** = Correlation is significant at the 0.01 level (2-tailed).

**Table 5 ijerph-22-00864-t005:** Predictors of acceptability, feasibility, and appropriateness of mobile phone messaging-based message-framing intervention, Jimma, Ethiopia (*N* = 379).

Dependent Variables	Independent Variables	Coeffs. (β)	SE	*p*-Value
Acceptability	Residence (Rural)	−1.827	0.596	0.002
	Wealth index			
	Poorest	1	1	1
	Poor	−0.366	0.640	0.567
	Medium	0.978	0.672	0.147
	Rich	2.193	0.851	0.010
	Richest	0.362	0.654	0.581
	UM messages receipt time			
	Morning	1	1	1
	Afternoon	−0.599	0.455	0.189
	Evening	−1.218	0.446	0.007
	Self-efficacy	0.354	0.053	0.000
	Engagement	0.202	0.059	0.001
Feasibility	Residence (Rural)	−1.885	0.493	0.000
	Education			
	Can not read or write	1	1	1
	Can read and write	1.470	0.617	0.018
	Primary school	0.970	1.261	0.442
	Secondary school	0.854	1.333	0.522
	Collage and above	0.542	1.501	0.718
	Occupation			
	Housewife	1	1	1
	Merchant	0.975	0.417	0.020
	Government employee	0.435	0.549	0.428
	Others ^a^	0.059	0.678	0.931
	Wealth index			
	Poorest	1	1	1
	Poor	0.311	0.530	0.557
	Medium	0.474	0.556	0.395
	Rich	1.439	0.704	0.042
	Richest	0.595	0.541	0.272
	UM message receipt time			
	Morning	1	1	1
	Afternoon	0.153	0.376	0.685
	Evening	−1.097	0.369	0.003
	Self-efficacy	0.132	0.044	0.003
	Engagement	0.251	0.048	0.000
	Satisfaction	0.183	0.052	0.001
Appropriateness	Residence (Rural)	−1.355	0.483	0.005
	Occupation			
	Housewife	1	1	1
	Merchant	3.727	0.409	0.000
	Government employee	4.086	0.538	0.000
	Others ^a^	2.477	0.664	0.000
	Satisfaction	0.256	0.051	0.000

^a^ (daily laborer and student).

## Data Availability

Data used for analysis in this study will be available upon reasonable request from the corresponding author.

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
