# Peer review of "Acceptability, Feasibility, and Appropriateness of Mobile Phone Messaging-Based Message-Framing Intervention for Promoting Maternal and Newborn Care Practices"

_ijerph, 2025, doi:10.3390/ijerph22060864_

Round 1
Reviewer 1 Report
Comments and Suggestions for Authors
Dear Author(s),
Thank you for your effort so far in getting your manuscript to review. The manuscript entitled ‘‘Acceptability, feasibility and appropriateness of mobile phone messaging-based message framing intervention for promoting maternal and newborn care practice’’. The subject of the manuscript is widely studied by other researchers. Nevertheless, I believe the work to have importance and I think it will be of use to other researchers. However, the structure of the paper needs to be improved. The similarity rate of the manuscript is very high and this should be taken into account and the necessary revisions should be made. It would be especially good if the similarity rate does not exceed 20%. I took quite some time deliberating on this manuscript. Ultimately, I recommend 'major revision' with encouragement to revise based on the extensive my comments from below:
1.Introduction
Line 69-70 “.....the evidence on the role of mobile phone messaging based message-framing interventions on maternal and neonatal health....” Please explain what “mobile phone messaging based message-framing” means. Why is it important to use this model? What are the limitations of the literature on this topic? And how will you fill this gap in the field? Essentially, the introduction should provide a background that answers all these questions.
Line 71-79: When I read this paragraph, I thought you were talking about your current research, not a previous research. It would be more appropriate to say “The study we conducted on X date...”. Please correct this paragraph.
Line 90-91: “.....However, there is limited evidence on these implementation outcomes for mobile phone messaging-based intervention.” You should explain what are the limitations mentioned here.
In general, the first three paragraphs of the introduction were very good (Line 1-68) emphasizing the importance of the topic. However, after line 69, I think it should be rewritten by paying attention to the points I mentioned at the beginning and research questions or hypotheses should be added.
- Materials and Methods
Information about the research design should be given before the study environment. The materials and methods section does not seem to bear the characteristics of scientific research. For example, while it is an experimental study consisting of a post-test, a cross-sectional design is mentioned. It would not be very appropriate to call the Study Design only cross-sectional. Because you are talking about an intervention. When I look at your other study you cited, I can understand that you call it a cross-sectional study, but the dates of your previous intervention studies are missing. How long after the intervention was this cross-sectional study conducted? This is not clear and you should base this time period on the literature. Again, how was the sample size determined? The population and sample should have been used instead of the study setting. And information about the study environment should have been written in this section.
Line 174-210: Cronbach's alpha coefficients for each scale should be written here. The values in Table 3 could have been included here. I do not think it is enough to write “...acceptable values of Cronbach's alpha coefficient.” below (line 216).
Line 213-214: “....Before the data collection, the tool was pretested on 5% of the sample size, and modifications were made to the questionnaire content based on the findings...” What were the changes mentioned here? Did these changes lead to changes in these coefficients?
Line 217-227: By which parameters did you decide whether the data were normally distributed or not? On what basis did you decide to use parametric methods? I do not find it appropriate to write the statistical analysis section, which is one of the most important points of a research, in such a passing manner. What should be written in detail?
- Results
Line 229-231: The following information should be deleted “This section may be divided by subheadings. It should provide a concise and precise description of the experimental results, their interpretation, as well as the experimental conclusions that can be drawn”
Line 233-234: “....a total of 379 mothers (187 from the gain-framed and 192 from the loss-framed group)...” Up to this point there was no mention of the definition of the two different groups mentioned above. I couldn't find them anywhere in the manuscript, please clarify. Also, if there are two separate groups in this way, comparisons should have been made in the tables as two separate groups (the gain-framed group and the loss-framed group), not over the whole participant.
In Table 1, the class boundaries in the Age variable are not equal, while there are 5 values between 15-19, 15 values between 20-34 and 10 values between 35-44, which does not give a correct result for comparison.
Table 1 shows that 20.3% of the participants are illiterate. However, you mention that your cell phone application contains text messages. How were you able to collect data from illiterate participants and is this valid? I understand that having an illiterate family member is not suitable for your intervention design, you were creating an intervention specific to the mother and the newborn.
- Discussion
The discussion section could have started by emphasizing which gap in the literature was closed with this research. In general, it would be more appropriate to examine the discussion one by one in terms of each variable addressed. It would also be nice to see more information on what the results of this research mean for the people of the region and how they will be reflected in future maternal and child health indicators.
Author Response
Reply to reviewers

Reviewer 2 Report
Comments and Suggestions for Authors
The introduction discusses the study - that comes at the end. The introduction should introduce the problem, not the study. I am not sure you need all of that data in the text in terms of the tables. What should the reader focus on? Make that stand out. The tools used could be better explained as well.
Comments on the Quality of English LanguageThere are some areas where the sentences are not complete and words are missing.
Author Response
Replay to reviewers

Reviewer 3 Report
Comments and Suggestions for Authors
The article entitled "Acceptability, Feasibility, and Appropriateness of a Mobile Phone Messaging-Based Message Framing Intervention for Promoting Maternal and Newborn Care Practices" was considered for review. The authors aimed to evaluate the acceptability, feasibility, and appropriateness of a mobile phone messaging-based message framing intervention. The research topic addressed in the manuscript is relevant to improving public health practices by utilizing contemporary technology that is increasingly integrated into daily life. However, some revisions are necessary, particularly in the Methods section. Below are additional comments:
Introduction: I recommend reviewing the phrasing related to the citation of the Randomized Controlled Trial. As currently written, it may imply that this is the primary objective of the study. I suggest rewording this section to clarify that the trial represents one stage within a broader study.
Methods: Regarding the message development phase, where the authors indicate that materials from the Ministry of Health and the World Health Organization were used, I suggest specifying the technique employed to select the content included in the messages (e.g., Content Analysis). Additionally, it is essential to clarify the process used to validate the messages, both by the target audience and by experts. It remains unclear whether a validation questionnaire was utilized. I also recommend specifying key dates, including the periods of recruitment, exposure, follow-up, and data collection. Furthermore, details on how the final study sample was determined should be provided, along with eligibility criteria, sources and methods of participant selection, and inclusion, exclusion, and discontinuation criteria. A more detailed description of the comparability of assessment methods between groups would also be beneficial.
Results: I suggest including the reasons for non-participation at each stage and, if feasible, incorporating a flow diagram to illustrate participant progression throughout the study.
Author Response
Reply to reviewers

Round 2
Reviewer 1 Report
Comments and Suggestions for Authors
Dear Author(s),
Thank you for taking into account and meticulously making all the necessary revisions. Your manuscript can be published as it is.